# Deglycosylation Increases the Aggregation and Angiogenic Properties of Mutant Tissue Inhibitor of Metalloproteinase 3 Protein: Implications for Sorsby Fundus Dystrophy

**DOI:** 10.3390/ijms232214231

**Published:** 2022-11-17

**Authors:** Jian Hua Qi, Bela Anand-Apte

**Affiliations:** 1Department of Ophthalmology, Cleveland Clinic Lerner College of Medicine, Cole Eye Institute, Cleveland Clinic Foundation, Cleveland, OH 44195, USA; 2Department of Molecular Medicine, Cleveland Clinic Lerner College of Medicine, Cole Eye Institute, Cleveland Clinic Foundation, Cleveland, OH 44195, USA

**Keywords:** angiogenesis, endothelial cell, glycosylation, matrix metalloproteinases, tissue inhibitor of matrix metalloproteinase, vascular endothelial growth factor (VEGF)

## Abstract

Sorsby fundus dystrophy (SFD) is an autosomal dominant macular disorder caused by mutations in tissue Inhibitor of the metalloproteinase-3 (*TIMP3)* gene with the onset of symptoms including choroidal neovascularization as early as the second decade of life. We have previously reported that wild-type TIMP3 is an endogenous angiogenesis inhibitor that inhibits Vascular Endothelial Growth Factor (VEGF)-mediated signaling in endothelial cells. In contrast, SFD-related S179C-TIMP3 when expressed in endothelial cells, does not have angiogenesis-inhibitory properties. To evaluate if this is a common feature of TIMP3 mutants associated with SFD, we examined and compared endothelial cells expressing S179C, Y191C and S204C TIMP3 mutants for their angiogenesis-inhibitory function. Western blot analysis, zymography and reverse zymography and migration assays were utilized to evaluate TIMP3 protein, Matrix Metalloproteinase (MMP) and MMP inhibitory activity, VEGF signaling and in vitro migration in endothelial cells expressing (VEGF receptor-2 (VEGFR-2) and wild-type TIMP3 or mutant-TIMP3. We demonstrate that mutant S179C, Y191C- and S204C-TIMP3 all show increased glycosylation and multimerization/aggregation of the TIMP3 protein. In addition, endothelial cells expressing TIMP3 mutations show increased angiogenic activities and elevated VEGFR-2. Removal of N-glycosylation by mutation of Asn^184^, the only potential N-glycosylation site in mutant TIMP3, resulted in increased aggregation of TIMP3, further upregulation of VEGFR-2, VEGF-induced phosphorylation of VEGFR2 and VEGF-mediated migration concomitant with reduced MMP inhibitory activity. These results suggest that even though mutant TIMP3 proteins are more glycosylated, post-translational deglycosylation may play a critical role in the aggregation of mutant TIMP3 and contribute to the pathogenesis of SFD. The identification of factors that might contribute to changes in the glycome of patients with SFD will be useful. Future studies will evaluate whether variations in the glycosylation of mutant TIMP3 proteins are contributing to the severity of the disease.

## 1. Introduction

Sorsby Fundus Dystrophy (SFD) is an autosomal dominant, fully penetrant, macular dystrophy with early onset of symptoms, usually in the third or fourth decade of life [1,2,3,4] and is caused by specific mutations in the Tissue Inhibitor of Metalloproteinase-3 (TIMP3) gene [5,6,7,8,9,10,11,12,13,14,15,16,17,18]. The majority of SFD mutations in TIMP3, result in the substitution of a cysteine residue for another amino acid in the C-terminus of the protein. Early symptoms encompass night blindness or a sudden decrease in visual acuity [1,5,19,20,21,22,23,24]. SFD closely resembles the exudative form of Age-Related Macular Degeneration (AMD) with Bruch’s membrane (BM) thickening, Photoreceptor-Retinal Pigment Epithelium-Choriocapillaris (PR-RPE-CC) atrophy and choroidal neovascularization (CNV) [5,20,25]. Large, confluent 20–30 μm thick, amorphous deposits between the basement membrane of the retinal pigment epithelium (RPE) and the inner collagenous layer of Bruch’s membrane are the predominant histopathological finding in SFD. These sub-retinal deposits in both SFD and AMD have been shown to be rich in TIMP3 [26,27,28,29].

TIMP3 protein is produced constitutively by the RPE and choroidal endothelial cells in the retina [29,30], is present in Bruch’s membrane [31], binds to sulfated glycosaminoglycans in the extra-cellular matrix (ECM) [32,33] and positioned strategically to regulate the choriocapillaris. While TIMP3 was originally identified as an inhibitor of matrix metalloproteinases (MMPs) [34,35], we identified it to be an inhibitor of tumorigenesis [36] as well as a potent angiogenesis inhibitor [37] mediating this function by blocking the binding of vascular endothelial growth factor (VEGF) to VEGFR2 [38]. This led to the hypothesis that mutant TIMP3 in SFD might be defective in its angiogenesis inhibition function and lead to choroidal neovascularization. We tested this hypothesis with the S179C (old nomenclature S156C) TIMP3 mutation and determined that expression of mutant TIMP3 in endothelial cells causes increased VEGF-mediated angiogenesis [39] which correlated with an increase in glycosylation of the mutant protein. Mice expressing S179C TIMP3 mutation also showed increased choroidal neovascularization in a laser-induced experimental model [40].

While an N-linked glycosylation motif (Asn^184^) in TIMP3 is conserved across all species [41], the functional consequence of glycosylation of TIMP3 is presently unclear. To determine the common molecular mechanisms of increased angiogenesis in SFD, we expressed three of the known SFD TIMP3 mutations (S179C-TIMP3, Y191C-TIMP3 and S204C-TIMP3) in endothelial cells to evaluate if the glycosylation state of the protein plays a role in the increased VEGF-mediated angiogenesis phenotype. 

## 2. Results

### 2.1. Increase in the Multimerized TIMP3 and Glycosylated Monomeric TIMP3 in the ECM of ECs Expressing SFD-Associated TIMP3 Mutations

To determine the effect of SFD-associated *TIMP3* mutations on VEGF-mediated angiogenesis we generated PAE/KDR cells (Porcine Aortic Endothelial cells with VEGFR2) expressing wild-type *TIMP3* (WT), *S179C-TIMP3*, *Y191C-TIMP3* or *S204C-TIMP3*. The ECM and conditioned medium (CM) from 2 to 3 independent clones from each stable transfection were analyzed by SDS-PAGE run under non-reducing and reducing conditions and Western blot analysis to detect the expression and molecular form of WT and mutant TIMP3. Under non-reducing conditions, WT-TIMP3 bound to the ECM in predominantly unglycosylated (24 kDa) form (Figure 1a, Lane 1-**) with a smaller fraction (also unglycosylated) being secreted into the CM (Figure 1b, Lane 1-**). However, mutant TIMP3 proteins (S179C, Y191C and S204C-TIMP3) in the ECM, multimerized (found at larger MWs at the top of the gel) with some proportion being secreted into the CM (Figure 1a,b, Lanes 2–7). Under reducing conditions, we observed that WT-TIMP3 binds to the ECM in predominantly unglycosylated form (24 kDa (**) Figure 1a’-Lane 1) with a smaller proportion being glycosylated (28 kDa-(*) Figure 1a’, Lane 1). In contrast mutant TIMP3 proteins, S179C-TIMP3 and Y191C-TIMP3 bound to the ECM, in a predominantly glycosylated form with a smaller proportion being unglycosylated (Figure 1a’-Lanes 2–7). The S204C-TIMP3 was present in the ECM as a doublet (27 kDa and 30 kDa) in equivalent proportions. The higher molecular weight bands of TIMP3 (approx. 200 kDa) seen under non-reducing conditions are reduced to monomeric TIMP3 in reducing gels. Quantitation of multimer vs. monomer of TIMP3 from at least 2 independent clones confirmed the preponderance of TIMP3 multimers in the ECM of endothelial cells expressing mutant TIMP3, particularly Y191C-TIMP3 (Figure 1f). However, the 48 kDa band cannot be reduced to monomeric form suggesting it to be an aggregate. Mutant TIMP3 forms more aggregates in the ECM) than wild-type TIMP3. WT-TIMP3 in endothelial cells is secreted into the conditioned medium (CM) predominantly as a glycosylated form (Figure 1b’-Lane 1) with a smaller proportion being unglycosylated (Figure 1b’-Lane 1). S179C-TIMP3 and Y191C-TIMP3 were secreted into the CM exclusively in a glycosylated form (Figure 1b’-Lanes 2–5). S204C-TIMP3 is also secreted into the CM of endothelial cells in a glycosylated form, but at significantly lower amounts (see inset of Figure 1b’). Quantitation of glycosylated to unglycosylated forms of TIMP3 from at least 2 independent clones confirmed the preponderance of glycosylated TIMP3 in the ECM of endothelial cells expressing mutant TIMP3 (Figure 1g). To confirm that the 28 kDa form of wild-type *TIMP3* (WT), *S179C-TIMP3* and *Y191C-TIMP3*, and the 30 kDa form of *S204C-TIMP3* is the glycosylated variant, we treated the ECM samples with PNGase F. PNGase F is an amidase that cleaves the innermost GlcNAc and asparagine residues of high mannose, hybrid and complex oligosaccharides making it a highly effective enzymatic method for removing almost all N-linked oligosaccharides from glycoproteins. Treatment of ECM (Figure 1c) and CM (Figure 1d) samples with PNGase resulted in almost complete loss of the higher 28 kDa glycosylated bands with a corresponding increase in the 24 kDa unglycosylated bands. As an added confirmation that the variants were glycosylated or unglycosylated forms of TIMP3, we treated the PAE cell lines with tunicamycin (TM), which inhibits N-glycosylation in eukaryotes by blocking the transfer of N-acetylglucosamine-1 phosphate (GlcNAc-1-P) from UDP-GlcNAc to dolichol-P (catalyzed by GlcNAc phosphotransferase; GPT). Treatment of PAE cells with Tunicamycin also resulted in a loss of the presumptive glycosylated forms of both wild-type and mutant TIMP3 (Figure 1e). Collectively, these results suggest that SFD-TIMP3 protein in endothelial cells binds to the ECM and is more glycosylated compared with the wild-type form of the protein which is present predominantly in the unglycosylated form. When present in high amounts (overexpression) wild-type TIMP3 is secreted into the CM of cells with an increased glycosylated to unglycosylated protein ratio, in contrast to the mutants S179C-TIMP3 and Y191C-TIMP3 which are secreted exclusively in the glycosylated form (Figure 1b). S204C-TIMP3 protein behaves a little differently in ECs compared with the other mutant TIMP3s, being present in the ECM as higher MW variants and with very little secreted into the conditioned medium.

### 2.2. Increase in Aggregated TIMP3 in the ECM and CM of ECs Expressing SFD TIMP3 Protein with N184Q Mutation at the Glycosylation Site

To determine the molecular and functional consequence of TIMP3 glycosylation, we generated mutant PAE/KDR lines in which we introduced wild-type *TIMP3* (WT), *S179C*-*TIMP3*, *Y191C*-*TIMP3* or *S204C*-*TIMP3* with an additional mutation at Asn^184^ (the only potential N-glycosylation site in TIMP3). The introduction of the N184Q mutation resulted in a complete shift of the 28 kDa form of WT-TIMP3 to a 24 kDa form in the ECM (Figure 2a’) and CM (Figure 2b’), seen in gels run under reducing conditions. In ECs expressing mutant *TIMP3* with N184Q mutation, the mutant TIMP3 protein was also present in the ECM in predominantly the unglycosylated form for *Y191C-TIMP3* and *S204C-TIMP3* mutants (Figure 2a’). Interestingly, the expression of S179C-TIMP3 with N184Q mutation resulted in an aggregated form (approximately 48 kDa and 72 kDa) in both ECM (Figure 2a’) as well as CM (Figure 2b’). This 48 kDa aggregated form was also present in increased amounts in the ECM of ECs expressing the *Y191C-TIMP3* and *S204C-TIMP3* (Figure 2a,b) mutations with N184Q when compared with TIMP3 protein expressing the same SFD mutations but with the glycosylation site intact (Figure 1a’,b’).

Under non-reducing conditions, WT-TIMP3 with N184Q mutation was found in both the ECM and CM in an unglycosylated 24 kDa form (**-Figure 2a,b, Lane1). In contrast, ECs expressing the SFD-TIMP3 mutations with N184Q are expressed as multimers in the ECM (Figure 2a, Lanes 2–7) and CM (Figure 2b, Lanes 2–7). These multimers are sensitive to reducing agents and are seen as monomers or aggregates under reducing conditions. Quantitation confirmed the preponderance of TIMP3 aggregate (Figure 2c) and multimer (Figure 2d) in the ECM of ECs expressing mutant TIMP3 with loss of the glycosylation site when compared with expression of mutant TIMP3 with the glycosylation site intact.

Collectively, these results confirm that Asn 184 is the major glycosylation site in TIMP3 and suggests that deglycosylation of mutant TIMP3 results in aggregation (Figure 2c) and multimerization (Figure 2d) of the protein.

### 2.3. Reduced MMP Inhibitory Activity in the ECM and CM of Cells Expressing SFD TIMP3 Protein

The ECM and CM from two independent PAE/KDR cell lines expressing WT or SFD-TIMP3 were subjected to reverse zymography for the determination of MMP inhibitory activity. In the ECM, the majority of the MMP inhibitory activity of WT-TIMP3 was seen in the unglycosylated variant (Figure 3a, lane 1*), while in the CM, both the glycosylated and unglycosylated WT-TIMP3 variants showed MMP inhibitory activity (Figure 3b, lane 1). In contrast, all three SFD mutants, S179CTIMP3, Y191CTIMP3 and S204CTIMP3 showed reduced MMP inhibitory activity of the glycosylated and unglycosylated forms in the ECM (Figure 3a lanes 2–7) as well as the CM (Figure 3b lanes 2–7). The secreted S179CTIMP3, Y191CTIMP3 and S204CTIMP3 while still attenuated in MMP inhibitory activity compared with WT-TIMP3, all showed some MMP inhibitory activity attributable to the glycosylated variant compared with the same protein/variant in the ECM. These results suggest that in ECs, expression of SFD mutant TIMP3 results in reduced MMP inhibitory activity.

### 2.4. Decrease in MMP Inhibitory Activity in the ECM and CM of ECs Expressing SFD TIMP3 Protein with N184Q Mutation at the Glycosylation Site

To test if glycosylation of TIMP3 modulates MMP inhibitory activity, we used reverse zymography to examine the MMP inhibitory activity of WT and SFD mutants in the presence of N184Q mutation. In ECs expressing *WT*-*TIMP3* with N184Q mutation, the TIMP3 protein showed reduced MMP inhibitory activity in both the ECM (Figure 3a, lane 8) and the CM (Figure 3b, lane 8) compared with WT-TIMP3. All three SFD mutants, S179CTIMP3, Y191CTIMP3 and S204CTIMP3 with N184Q mutation showed an absence of MMP inhibitory activity in the CM (Figure 3b, lanes 9–14). Interestingly, the aggregated form of S179C TIMP3 with the N184Q mutation showed some MMP inhibitory activity both in the ECM (Figure 3a, lane 9,10) and CM (Figure 3b, lane 9,10). Collectively, these results suggest that the majority of MMP inhibitory activity of mutant TIMP3 in ECs is in the secreted glycosylated form (albeit markedly lower than wild-type) and mutation of the glycosylation site on the protein results in further reduction in the MMP-inhibitory activity of the protein (Figure 3d,e) in ECM and CM, respectively.

### 2.5. N184Q Mutation in the TIMP3 Gene Results in Increased Secreted MMP 2 and MMP9 Activity in ECs

To determine the consequence of reduced MMP inhibitory activity of TIMP3 due to SFD mutation, the CM from 2 independent PAE/KDR cell lines expressing wild-type *TIMP3* (W) (Figure 3c-lane 1), *S179C-TIMP3* (Figure 3c-lane 2), *Y191C-TIMP3* (Figure 3c-lane 3) or *S204C-TIMP3* (Figure 3c-lane 4,5) were subjected to gelatin zymography to detect gelatinase (MMP2 and MMP9) activity. S179C, Y191C- and S204C-*TIMP3* expression led to modest increases in proMMP-9 (92 kDa) and active MMP-9 (88 kDa) as well as active MMP2 (68 kDa) relative to WT-TIMP3 (Figure 3c lanes 4–6). These results suggest that SFD-TIMP3 may upregulate MMP2 and MMP9 activity because of decreased MMP inhibitory activity (Figure 3f,g).

To determine if TIMP3 glycosylation modulates SFD-TIMP3-driven MMP activity, we examined the effect of N184Q mutation on MMP activity in the CM of ECs. Zymograms of the CM from cells expressing N184Q-WT-TIMP3 revealed significantly more active MMP-2 and MMP-9 forms than that from WT-TIMP3 control (Figure 3c-lane 6,7). Similarly, the CM from cells expressing N184Q-S179C, N184Q-Y191C-TIMP3 and N184Q-S204C revealed more active MMP-2 and MMP-9 activity (compared with those from the corresponding S179C-TIMP3, Y191C-TIMP3, S204C-TIMP3 or N184Q-WT-TIMP3 control (Figure 3f,g). These results suggest that mutation at N184Q which inhibits glycosylation of TIMP3 results in an increase in MMP activity in ECs.

### 2.6. SFD-Associated TIMP3 Mutations Induce An Upregulation of VEGFR-2 in ECs That is Modulated by the Glycosylation of TIMP3

We have previously reported that S179C-TIMP3 when expressed in ECs results in increased VEGFR-2 protein on the surface [39]. To determine if the upregulation of VEGFR2 on the surface of ECs was also induced by other SFD-TIMP3 mutants, we evaluated lysates from PAE/KDR lines expressing WT, S179C, Y191C and S204C-TIMP3, which were subjected to Western blotting to detect VEGFR-2 protein. VEGFR2 in these cells is generally present in mature (210 kDa), intermediate (200 kDa) and immature (160 kDa) forms as reported previously [39]. As we have previously demonstrated, the expression of S179C-TIMP3 increased the level of 210 and 200 kDa forms of VEGFR2 on the surface of ECs when compared with ECs expressing *WT-TIMP3* (Figure 4a, lane 7, 3). We now demonstrate that Y191C- (Figure 4a, lane 11) and S204C-TIMP3 (Figure 4a, lanes 19,23) also exhibit a similar upregulation of VEGFR-2. These results suggest that SFD mutations in TIMP3 induce the upregulation of VEGFR-2 in ECs (Figure 4d).

To determine if TIMP3 glycosylation is involved in SFD-TIMP3-driven upregulation of VEGFR-2, we evaluated the effects of N184Q mutation on the ability of SFD mutations to induce upregulation of VEGFR2 on ECs. We compared the levels of VEGFR2 on the surface of cells expressing WT/N184Q, S179C/N184Q, Y191C/N184Q and S204C/N184Q with cells expressing wild-type TIMP3 and SFD mutant TIMP3 with the glycosylation site intact. The N184Q mutation in wild-type TIMP3 resulted in a decrease in the expression of VEGFR2 on ECs (Figure 4a, lane 5 and Figure 4e). Surprisingly, the introduction of N184Q mutation in S179C (Figure 4a, lane 9, Figure 4f), Y191C (Figure 4a, lane 13, Figure 4g) and S204C-TIMP3 (Figure 4a, lane 21, 25, Figure 4h) led to increased 210 and 200 kDa forms of VEGFR-2 in ECs compared with cells expressing corresponding SFD mutations with an intact glycosylation site (Figure 4f–h). Actin was used as a loading control (Figure 4c).

### 2.7. SFD-Associated TIMP3 Mutations Result in Enhanced VEGFR2 Autophosphorylation in Response to VEGF That Is Modulated by the Glycosylation of Mutant TIMP3

In our previous work, we determined that ECs expressing S179C TIMP3 had increased autophosphorylation of VEGFR2 in response to VEGF when compared with ECs expressing wild-type TIMP3. To determine if other TIMP3 mutations associated with SFD could have the same effect, we evaluated VEGFR2 phosphorylation in ECs expressing WT, S179C, Y191C and S204C-TIMP3 following a 5 min exposure to VEGF (Figure 4b). Consistent with our previous results with S179C [39], expression of all mutant forms increased VEGF-stimulated VEGFR2 auto-phosphorylation relative to that seen in ECs expressing wild-type TIMP3 (Figure 4b,i). To determine if mutant glycosylation modulates VEGFR-2 signaling, lysates from cells expressing N184Q-WT or N184Q-SFD-TIMP3 were analyzed similarly following stimulation with VEGF. The N184Q mutation in wild-type TIMP3 did not increase VEGFR2 autophosphorylation. In fact, N184Q mutation in WT-TIMP3 resulted in reduced VEGFR2 expression as well as reduced VEGFR2 phosphorylation following exposure to VEGF (Figure 4b, lane 6 and Figure 4j). However, mutation of the glycosylation site in all SFD-TIMP3 mutants S179C- Y191C- and S204C-TIMP3 resulted in increased VEGF-stimulated VEGFR-2 autophosphorylation when compared with the corresponding mutants without N184Q mutation (Figure 4b,k–m). This suggests that glycosylation of mutant TIMP3 may regulate VEGF-VEGFR2 signaling in ECs.

### 2.8. SFD-Associated TIMP3 Mutations Result in Increased VEGF-Stimulated Migration of ECs That Is Modulated by Glycosylation of TIMP3

To determine if elevated VEGFR-2 signaling by SFD-TIMP3 could lead to increased biological effects of VEGF, we investigated if SFD-TIMP3 mutants induce increased migration of ECs cells using a mini-Boyden chamber chemotaxis assay. Expression of wild-type and mutant TIMP3 (with and without mutation of the glycosylation site) did not change the baseline migration properties of ECs (Figure 5a). To determine if increased VEGF-mediated migration was induced by SFD-TIMP3 mutants, we evaluated VEGF-induced chemotaxis in ECs expressing WT, S179C, Y191C and S204C-TIMP3. Following VEGF stimulation, S179C-, Y191C- and S204C showed increased migration relative to ECs expressing wild-type TIMP3 (Figure 5b). To evaluate if the glycosylation status of mutant TIMP3 could regulate the VEGF-mediated migration of ECs we performed these experiments in cells expressing WT or SFD mutations with the N184Q mutation at the glycosylation site. While mutation of the glycosylation site in wild-type TIMP3 further reduces the migratory phenotype in ECs, the N184Q mutation in S179C, Y191C and S204C resulted in increased chemotaxis towards VEGF (Figure 5c).

## 3. Discussion

Glycosylation is one of the predominant post-translational modifications that generate a functional diversity in proteins. Alterations in the glycan structures of proteins have been demonstrated to modify the conformational stability of proteins due to modifications in their physical and chemical properties leading to alterations in functions that contribute to the development and progression of several diseases [42]. The four-member family of tissue inhibitors of metalloproteinases (TIMPs) was originally identified as proteins that serve as physiological inhibitors of matrix metalloproteinases (MMPs). The homeostasis of the extracellular matrix (ECM) is controlled by a balance between MMPs and TIMPs and disruption in this tightly regulated process can lead to ECM breakdown or accumulation. While glycosylation is a post-translational modification found in MMPs, the implications of this on ECM regulation have not yet been explored [43]. Of the four human TIMPs, N-linked glycans have been found only in TIMP1 and 3. NetNGyc and NetOGlyc tools [44] confirm that hTIMP3 protein has a single glycosylation site (Asn 184) near the carboxy terminus and no predicted O-glycosylation sites. Interestingly, the extent of glycosylation of TIMP3 appears to be cell-type specific. Our previous studies suggest that TIMP3 in RPE cells in culture is predominantly unglycosylated [40] but in endothelial cells (as seen in this study), TIMP3 is expressed in both glycosylated and unglycosylated forms. In addition, a few recent studies have evaluated the role of TIMP3 mutations on RPE function using human iPSC-derived RPE cells [45,46]. One of these explores the interesting hypothesis that excess TIMP3 rather than dysfunctional TIMP3 in RPE cells drives the pathology in SFD [45].

It has been previously postulated that the function of TIMP3 might also be unique to specific cell types. In the present study, using endothelial cells, we evaluated the role of glycosylation on the function of both wild-type and SFD mutant TIMP3 protein. Our initial observations determined that wild-type TIMP3 in the ECM of endothelial cells was predominantly the unglycosylated form compared with both glycosylated and unglycosylated forms being secreted out of the cell in the CM. In contrast, the three TIMP3 mutants appeared to be present in the ECM in predominantly the glycosylated form. Mutant TIMP3 showed reduced MMP inhibitory activity in ECs but disruption of the single glycosylation site resulted in a further decrease in MMP inhibitory activity in both wild-type and mutant TIMP3. This suggests that glycosylation of TIMP3 is essential for effective MMP inhibition. However, it appears not to be the only factor contributing to MMP inhibition, as SFD mutant TIMP3, while more glycosylated shows reduced MMP inhibitory activity compared with the wild-type protein. The effect of deglycosylation of mutant TIMP3 on EC apoptosis has yet to be determined [47].

It should be noted that the changes in the glycosylation and aggregation of mutant TIMP3, although always more than the wild-type protein, vary in amount between the different mutations tested. This may be a consequence of different amounts of mutant protein being expressed in cells. Alternatively, it is possible that the introduction of a new cysteine residue in mutant TIMP3 can result in altered disulfide bonds which could lead to varying degrees of abnormal folding or altered conformation of the protein, exposure of the glycosylation site and increased attachment of glycans and/or susceptibility to glycosidases. Our results reported here suggest that S179C and Y191C mutants in endothelial cells, exhibit more glycosylation than the S204C mutant protein. This could be due to the fact that the unpaired cysteine in S204C is in close proximity to the glycosylation site, Asn^184^ which could interfere with exposure of Asn184 leading to reduced glycosylation.

Our results also indicate that deglycosylation of TIMP3 mutant proteins leads to the formation of TIMP3 aggregates (seen in gels run under reducing conditions and differentiates them from multimers). This is especially severe in S179C mutants that lack the N184Q glycosylation site. We hypothesize that loss of glycosylation may increase the sensitivity of altered disulfide bonds to oxidation, leading to oxidative crosslinking. Clinical studies suggest that the age of onset of symptoms in SFD is earlier in patients with cysteine substitutions at amino acids 179 and 191 (2nd–3rd decade of life) compared with patients with mutations at Ser204 who show the onset of symptoms at a later stage (3rd-6th decade of life). It is possible that loss of glycosylation and changes in protein aggregation may lead to more severe alterations in TIMP3 structure and loss of function (angiogenesis and MMP inhibition) and early onset and more severe symptoms. Although the loss of glycosylation of TIMP3 is not a known molecular consequence of SFD-related mutation, ‘N-glycosylation/de-glycosylation systems may be a biologically important mechanism of post-translational re-modification of certain proteins [48,49] and in the case of SFD-deglycosylation due to as yet unknown factors may accelerate the progression of the disease. Identifying the factors or mechanisms of deglycosylation in SFD patients with severe phenotypes may lead to the identification of novel therapeutic approaches for SFD.

Our previous studies suggested that S179C TIMP3 expression in ECs resulted in increased expression of VEGFR2 on the surface of ECs which resulted in increased VEGF-mediated autophosphorylation of VEGFR2 and downstream signaling [39]. In this study, we confirm that in addition to S179C TIMP3, mutants Y191C and S204C also result in increased expression of VEGFR2 on the surface of ECs as well as increased VEGF-induced autophosphorylation and downstream angiogenic activity (VEGF-induced migration). Using the N184Q mutation that results in loss of glycosylation, we observed a further increase in VEGFR2 and VEGF-mediated signaling in cells expressing mutant TIMP3 but a reduced expression of VEGFR2 in cells expressing wild-type TIMP3. These data suggest that glycosylation has opposing effects on wild-type and mutant TIMP3 with respect to signaling and angiogenesis. Of interest, is a recent study that demonstrates that inhibition of protein glycosylation by a hexosamine, D-mannosamine stimulates and accentuates VEGF-mediated angiogenesis [50]. Previous studies had suggested that inhibiting glycosylation may result in angiogenesis inhibition [51]. Our study demonstrates that loss of glycosylation has opposing effects on wild-type TIMP3 and mutant TIMP3 with respect to angiogenesis. Loss of glycosylation of all three mutant TIMP3 proteins resulted in increased VEGFR2 expression as well as VEGF-mediated signaling and angiogenesis. No effect of deglycosylation was seen in wild-type TIMP3. Future studies will evaluate the molecular mechanisms that result in these divergent effects of glycosylation with wild-type and mutant TIMP3.

Accumulation of floccular deposits is a pathological feature in SFD [52]. It is of interest to note a significant increase in multimerized as well as aggregated TIMP3 in the ECM and CM of ECs expressing SFD-mutant TIMP lacking the glycosylation site. It is tempting to speculate that the glycosylation machinery may play an important role in the production of aggregated protein/deposits and might be a potential target for therapeutics to slow down the progression of the disease. Recent studies have evaluated the potential utility of recombinant TIMP3 in cardiovascular diseases [53] and engineered TIMP3 molecules with an additional glycosylation site led to improved expression and cardiac function in a rodent model of myocardial infarction [54]. However, our results suggest that deglycosylation results in increased multimerization/aggregation of TIMP3 in endothelial cells. Evidence for the role of glycosylation in regulating tau as well as amyloid-b-precursor protein has been recently demonstrated [55,56]. In addition, glycosylation has been shown to inhibit the aggregation of human prion protein and decrease its toxicity [57].

These findings raise some significant questions regarding the biology of TIMP3 and the molecular mechanisms that regulate its localization and function in endothelial cells in addition to post-translational modifications such as deglycosylation that could lend further impact. A better understanding of the structure and function of wild-type and mutant TIMP3 could provide important clues to the pathogenesis of Sorsby Fundus Dystrophy as well as the more common and related age-related macular degeneration.

## 4. Materials and Methods

### 4.1. Materials

Porcine Aortic Endothelial (PAE and PAE_KDR_) [38,39] cells that ectopically express VEGFR2 (KDR) were cultured in Ham’s F-12/DMEM medium supplemented with 10% fetal calf serum (FBS) (Cambrex, Fisher Scientific, Dallas, TX, USA), 50 units/mL penicillin and 50 μg/mL streptomycin as described previously [38,39]. For most experiments, we used recombinant human VEGF (R&D Systems, Minneapolis, MN, USA), PNGase F (New England Biolabs, Ipswich, MA, USA), and tunicamycin (Sigma, St Louis, MO, USA). For Western Blot analysis we utilized the following antibodies: monoclonal anti-TIMP-3 antibody (Chemicon International, Inc., Temecula, CA, USA), monoclonal anti-Flk-1 (Santacruz, sc-6251 and A-3), b-actin (SantaCruz Biotechnology, Dallas, TX, USA), phospho-VEGF receptor-2 (Cell Signalling Technology, Danvers, MA, USA).

### 4.2. Construction of TIMP3 Mutants and Cellular Transfection

A single point mutation at Asn^184^ of WT-, S179C, Y191C and S204C-TIMP3 in which Asn was changed to a Gln (N184Q-WT-, N184Q-S179C, N184Q-Y191C, and N184Q-S204C-TIMP3) were constructed using Quick-Change II XL Site-Directed Mutagenesis Kit. The sequences of the PCR primer pairs for N184Q-WT, N184Q- S179C and N184Q-Y191C -TIMP3 mutants were sense GAT AAA AGC ATC CAA GCC ACA GAC CCC and anti-sense GGG GTC TGT GGC TTG GAT GCT TTT ATC. For N184Q-S204C-TIMP3 were sense GAT AAA TGC ATC CAA GCC ACA GAC CCC and anti-sense GGG GTC TGT GGC TTG GAT GAT GCA TTT ATC. All TIMP3 mutations were verified by DNA sequencing. The sequence of PCR primer used was: pCEP forward 5′-AGC AGA GCT CGT TTA GTG AAC CG-3′. The expression constructs were transfected stably into PAE/KDR cells using Effectene® Transfection Reagent (QIAGEN, Hilden, Germany) according to the manufacturer’s instructions. Stable clones were isolated using *Hygromycin B* selection. Expression of WT-and mutant TIMP3 was identified by immunoblotting with anti-TIMP3 antibody. 

### 4.3. Preparation of ECM and Conditioned Media

A 2-day serum-free conditioned medium (CM) was collected from sub-confluent cells cultured on six-well plates (Corning Costar, Corning, NY, USA). The cell monolayer was dislodged from the culture plates following a 10 min incubation in Ca^2+^, Mg^2+^ free phosphate-buffered saline (PBS) containing 2.5 mM EDTA. After several rinses in PBS and water, the ECM was scraped in a small volume of electrophoresis sample buffer without a reducing agent for analyses. 

### 4.4. Immunoblotting

The cells were lysed in lysis buffer composed of 20 mM Tris-HCl, pH 7.5, 150 mM NaCl, 10% glycerol, 1% Triton X-100, 0.5 mM Na_3_VO_4_ and protease inhibitor cocktail tablets (Roche, Basel, Switzerland). The concentrations of proteins in lysates were measured using the Pierce BCA protein Assay kit (Thermo Scientific, Waltham, MA, USA). Equal amounts of proteins were heated in sample buffer, separated by SDS-PAGE and electrophoretically transferred onto PVDF membrane (LI-COR). The filter was incubated in Odyssey Blocking buffer at room temperature for 1 h and probed with specific antibodies overnight at 4 °C. After washing, the filter was incubated with Dilute IRDye secondary antibody (LI-COR) followed by an Odyssey Family Imaging System. The blots were re-probed after the removal of the first probe by incubation in NewBlot PVDF 1× stripping buffer (LI-COR, Lincoln, NE, USA) for 20 min at Room temperature. 

In some cases, proteins were probed with antibodies and detected with either an HRP-conjugated anti-rabbit or anti-mouse IgG antibody (Amersham Pharmacia Biotech., Piscataway, NJ, USA) followed by ECL. The blots were stripped with Western ReProbe^TM^ solution (Genotech, St Louis, MO, USA) for 30 min and re-probed as indicated. 

### 4.5. Zymography and Reverse Zymography

Equal amounts of non-reduced samples were loaded onto a 7.5% gel with 1 mg/mL gelatin (zymography) or onto a 12% gel plus the conditioned media from retinal pigment epithelial cells (ARPE-19) cells treated with 100 nM PMA, as a source of MMPs for reverse zymography. Following electrophoresis, gels were processed as described previously [29,38,40,47]. Briefly, gels were agitated in a solution of 25 mg/mL Triton X-100 to remove SDS. The Triton X-100 was washed off with water, and the gels were then incubated for 16 h in 50 mM Tris-HCl (pH 7.5) containing 5 mM CaCl_2_ and 0.2 mg/mL sodium azide at 37 °C. Gels were stained with 5 mg/mL Coomassie Blue R-250 in acetic acid/methanol/water (1:3:6) for 1–2 h and de-stained with acetic acid/methanol/water (1:3:6). 

### 4.6. Migration Assay

A modified Boyden chamber assay was carried out as described previously. Briefly, 8.0 μm pore PVPF polycarbonate membranes were pre-coated with 100 μg/mL collagen type I in 0.2 N acetic acid (Cohesion technologies, INC., Palo Alto, CA, USA). VEGF at the indicated concentrations was placed in the lower chamber, and cells (2 × 10^6^) in serum-free medium were placed in the upper chambers. Chamber was then incubated for 4 h at 37 °C in a 5% CO_2_ humidified incubator. Cells remaining on the top of the filter were removed. Cells on the bottom surface of the filter were fixed, stained and mounted. The number of migrating cells per well was counted microscopically.

### 4.7. Statistical Analysis

Data points in each group were evaluated for normality on GraphPad Prism and were found to be distributed normally. Data are expressed as mean ± SEM and analyzed and graphed using GraphPad Prism (Prism 9 for MacOS version 9.4.1). Analysis of Variance (ANOVA) was used to quantitate differences between means. A *p*-value < 0.05 was considered significant.

## Figures and Tables

**Figure 1 ijms-23-14231-f001:**
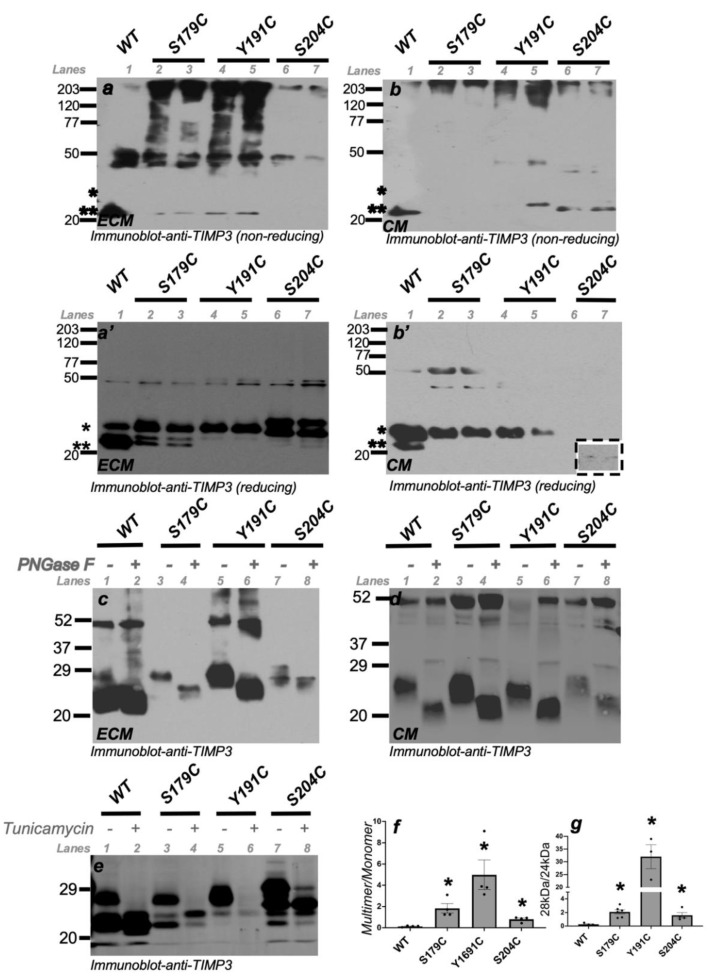
Glycosylated and unglycosylated forms of TIMP3 in ECM (extracellular matrix) and CM (conditioned medium) of ECs. ECM (**a**,**a’**,**c**,**e**) and CM (**b**,**b’**,**d**) were collected from ECs expressing wild-type TIMP3 (W), and SFD mutations, S179C TIMP3, Y191C and S204C. ECM and CM were subjected to non-reducing (**a**,**b**) and reducing (**a’**, **b’**,**c**–**e**) SDS-PAGE and Western blot analysis for TIMP3 protein expression. * indicates glycosylated TIMP3, ** indicates unglycosylated TIMP3. Glycosylated forms were confirmed following treatment of ECM preps with PNGase F (**c**,**d**) and cells with Tunicamycin (**e**). Ratio of Multimer to monomer TIMP3 (**f**) and glycosylated to unglycosylated TIMP3 (**g**) was quantitated from Western blot images with Image J. *n* ≥ 3. * *p* ≤ 0.002.

**Figure 2 ijms-23-14231-f002:**
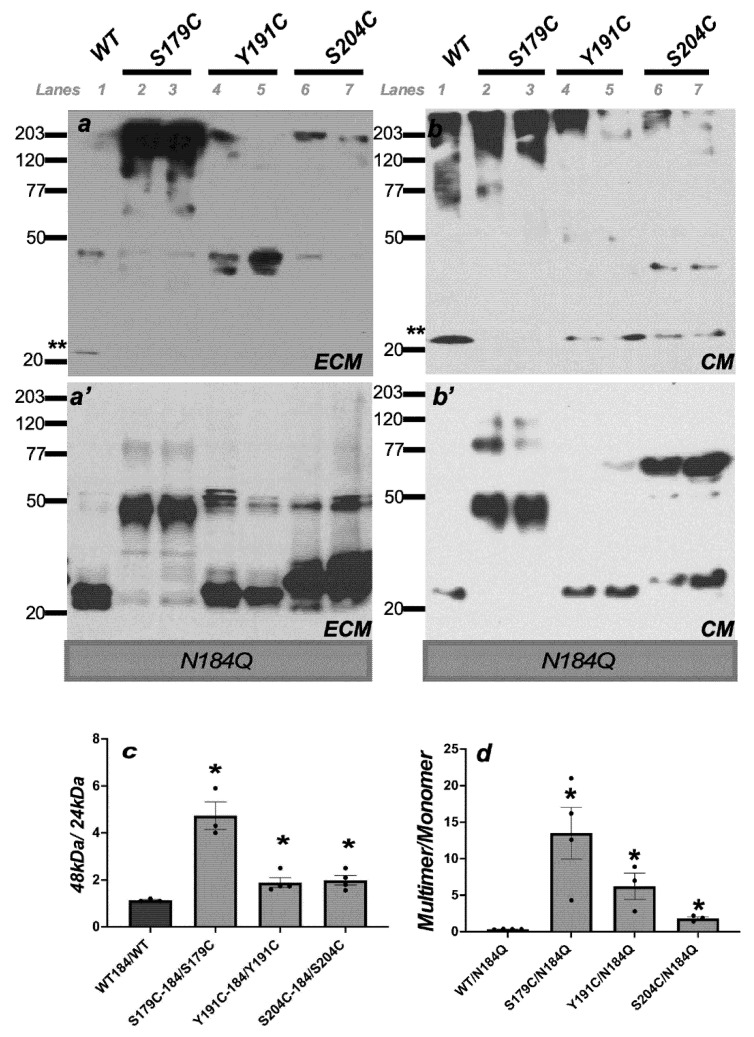
N184 is the major glycosylation site in TIMP3. ECM (extracellular matrix) (**a**) and CM (conditioned medium) (**b**) were collected from ECs expressing wild-type TIMP3 (W), and SFD mutations, S179C TIMP3, Y191C and S204C with an additional N184Q glycosylation mutation. ECM and CM were subjected to non-reducing (**a**,**b**) and reducing (**a’**,**b’**) SDS-PAGE and Western blot analysis for TIMP3 protein expression. ** indicates unglycosylated TIMP3. Ratio of aggregate (48 kDa) TIMP3 to unglycosylated 24 kDa TIMP3 (**c**) and multimer to monomer TIMP3 (**d**) was quantitated from Western blot images with Image J. *n* ≥ 3. * *p* ≤ 0.005.

**Figure 3 ijms-23-14231-f003:**
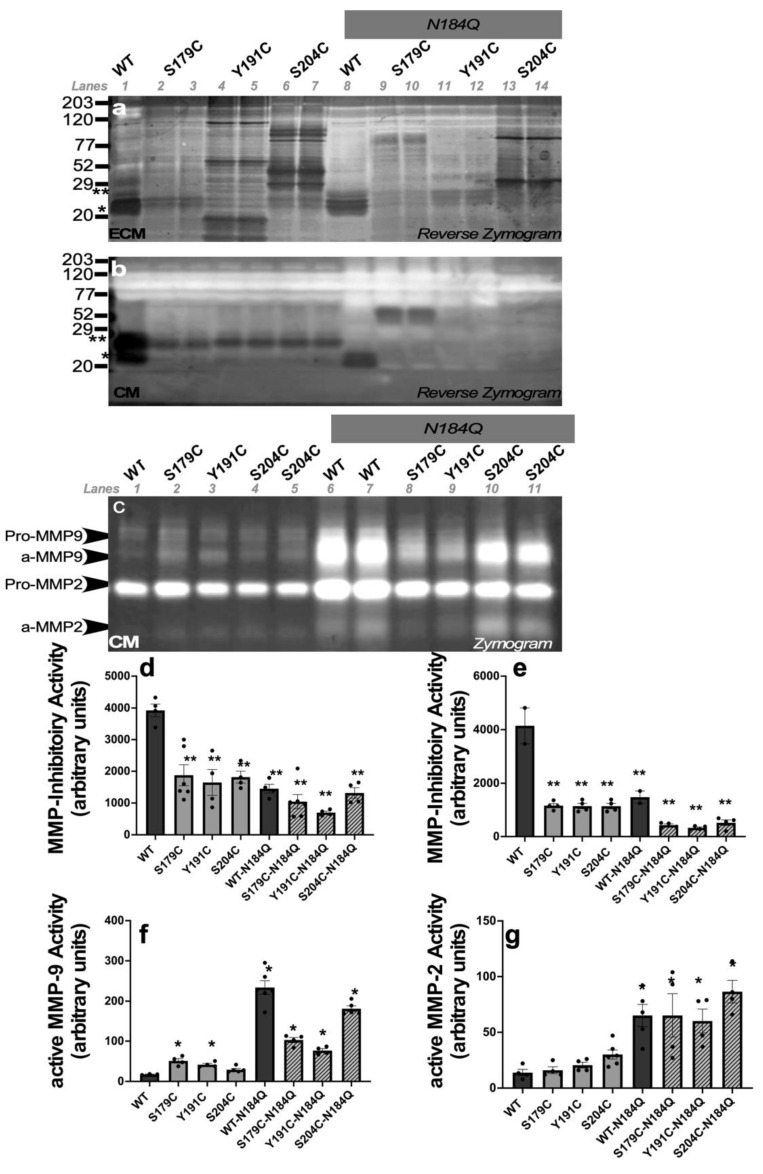
TIMP and MMP activity of ECM and CM of ECs expressing wild-type and mutant TIMP3. ECM (**a**) and CM (**b**) were collected from ECs expressing wild-type TIMP3 (W), and SFD mutations, S179C TIMP3, Y191C and S204C with and without N184Q glycosylation mutation. * indicates glycosylated TIMP3, ** indicates unglycosylated TIMP3. ECM and CM were subjected to reverse zymogram analysis for TIMP3 activity and CM was subjected to zymography (**c**) for MMP activity analysis. Quantitation of MMP inhibitory activity in ECM (**d**) and CM (**e**), active MMP-9 (**f**) and active MMP-2 (**g**). *n* ≥ 3, **p* < 0.04, ** *p* < 0.0001.

**Figure 4 ijms-23-14231-f004:**
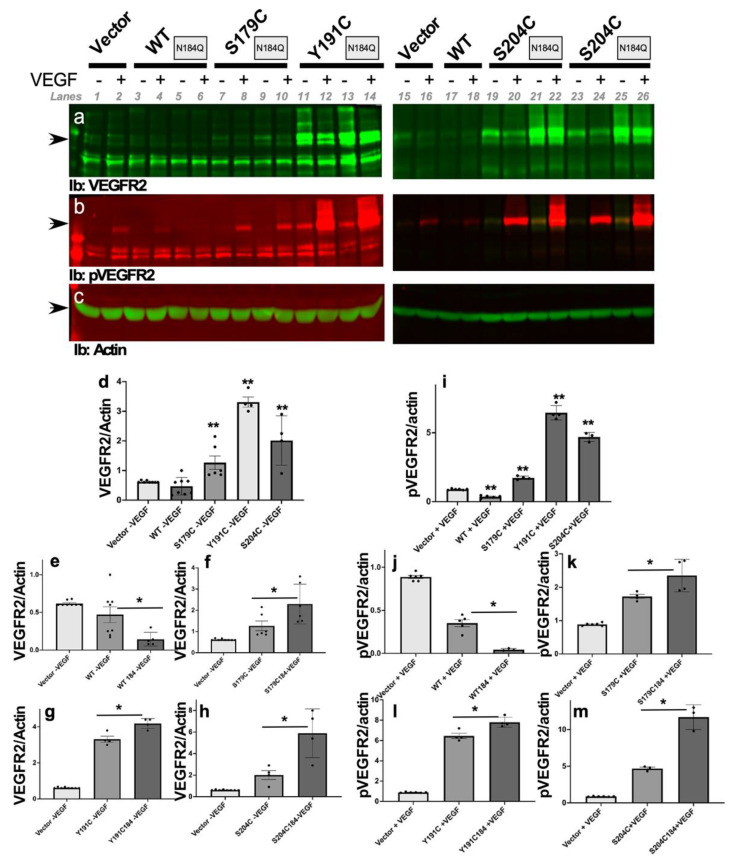
Increased VEGFR2 and VEGF-induced phosphorylation of VEGFR2 in ECs expressing wild-type and mutant TIMP3. Cell lysates were prepared from ECs expressing Empty Vector (Vector), wild-type TIMP3 (WT), and SFD mutations, S179C TIMP3, Y191C and S204C with and without N184Q glycosylation mutation following treatment with VEGF for 5 min. Lysates were subjected to SDS-PAGE and western bot analysis and probed with antibodies against VEGFR2 (**a**) pVEGFR2 (**b**) and Actin (**c**). Quantitation of the ratio of VEGFR2: Actin (**d**–**h**) and pVEGFR2: Actin (**i**–**m**) was performed using densitometry of bands in Image Studio (*n* ≥ 3). * *p* < 0.027, ** *p* < 0.0001.

**Figure 5 ijms-23-14231-f005:**
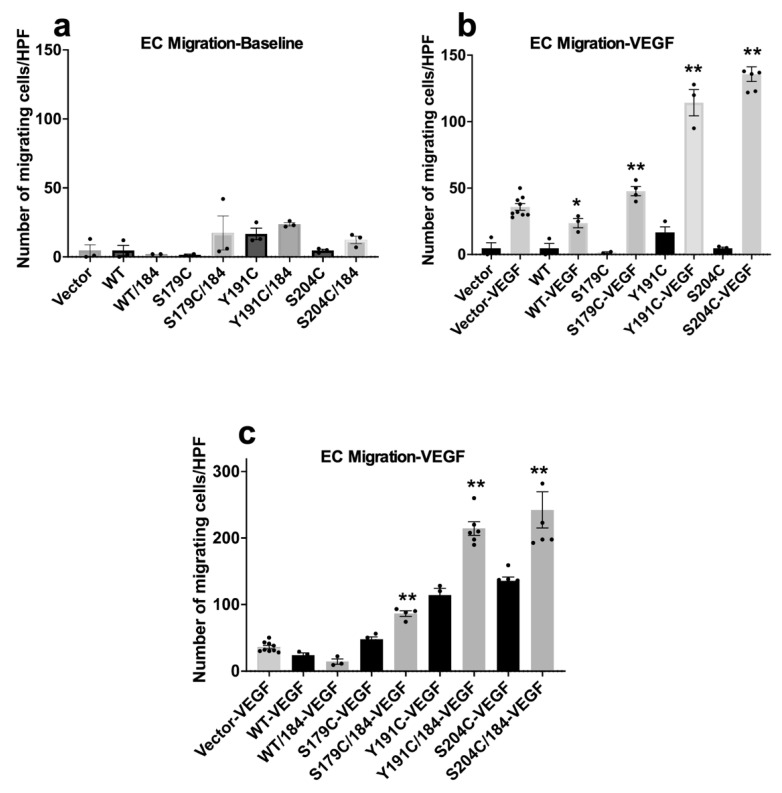
Mutant TIMP3 is an inefficient inhibitor of VEGF-mediated EC migration Serum-starved PAE_KDR_ cells expressing wild-type (WT) and SFD mutations, S179C TIMP3, Y191C and S204C with and without N184Q glycosylation mutation were evaluated for base-line migration (**a**) and VEGF (50 ng/mL)-induced migration (**b**,**c**). Number of migrating cells/HPF is expressed as means ± SEM of quadruplicate samples. *n* ≥ 3, * *p* < 0.001, ** *p* < 0.0001.

## Data Availability

All data are contained within the manuscript.

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
