# Peer review of "Deglycosylation Increases the Aggregation and Angiogenic Properties of Mutant Tissue Inhibitor of Metalloproteinase 3 Protein: Implications for Sorsby Fundus Dystrophy"

_ijms, 2022, doi:10.3390/ijms232214231_

Round 1

Reviewer 1 Report

Resumen

Este documento tiene un gran potencial, está bien organizado y utiliza figuras, imágenes y gráficos útiles para resumir rápidamente los resultados. Aunque este documento debe abordar una serie de cuestiones importantes a mejorar principalmente por su limitación en la estructura como artículo científico Comentarios específicos

·        Resumen

Los autores deben considerar mejorar los criterios de resumen. Deben incluir la implicación clínica y añadir el alcance de las conclusiones.

·        Introducciones

Line 34-48.  Authors should rewrite part of the introduction as they incur a copy of the 2019 study abstract ``Anand-Apte, B., Chao, J. R., Singh, R., & Stöhr, H. (2019). Sorsby fundus dystrophy: Insights from the past and looking to the future. Journal of neuroscience research97(1), 88–97. https://doi.org/10.1002/jnr.24317``  although it is a personal contribution of the same authors of previous works, the authors are incurring in an ethical fraud or plagiarism of these lines already written for another scientific study.

·       Methodology

The section on materials and methods is the most important part of a scientific study. The authors need to improve the heterogeneity of the study design. You must use a section to mention the statistical methods used for calculations, regularities, patterns, mathematical trends or classification of the types of random variables that make up the problematization of the research.

·       Conclusions

The conclusions refer to the summary expression of the results obtained. Proposing main findings and hypotheses of the research. The authors must make a section of conclusions to generate arguments and statements.

·       References

La referencia bibliográfica se utiliza en un artículo científico para que los autores puedan fundamentar, mantener y respaldar sus afirmaciones. Sin embargo, el apartado de referencias muestra un exceso de trabajo científico poco actual. Los autores deben actualizar y fundamentar sus referencias con obras de mayor actualidad existente.

Reviewer 2 Report

The experimental findings described by Qi and Anand-Apte (ijms-1894545) extend previously published work by this group using porcine aortic endothelial cells overexpressing TIMP3 and SFD-related mutants. The main goal of the study is to shed light on the still unsolved pathomechanism of Sorsby fundus dystrophy.

I have two major points of criticism that both relate to the interpretation of the results:

1. The authors use three different SFD-related mutants in their experiments and propose common features. However, the data show significant differences among the mutants in almost all of the properties tested (for instance: S204C proteins aggregate to a much lesser extent and show a different pattern of glycosylation, S179C-N184Q forms aggregates under reduced conditions, S204C-N184Q is secreted as a higher molecular complex). These differences need to be further addressed.

2. This and previous studies have shown that SFD-related mutations cause an increase in glycosylation/aggregation of TIMP3 proteins and angiogenic activities. It therefore seems paradoxical that deglycosylation further increases these features. Since loss of glycosylation of TIMP3 is not a known molecular consequence of SFD-related mutations, direct implications of the data obtained with N184Q mutagenesis for SFD pathology are not visible. Further clarification is needed and the text needs to be adjusted to avoid overinterpretation (title etc.)

Minor points:

1. The onset of symptoms in SFD are highly variable and typically start > 30 y. It is misleading to describe SFD as a disease with early onset.

2. The lanes in the figures should be numbered when numbers are given in the text.

3. Glycosylated and un-glycosylated forms of TIMP3 should be marked consistently with one or two asterisks. Similarly, mutants harboring the N184Q mutations should be named consistently in the figures.

4. Results, section 2.6/2.7. Transfected ECs with empty vector show a higher upregulation of VEGFR2 and more pronounced, an increased VEGFR2 phosphorylation than ECs overexpressing WT-TIMP3 (Fig. 4 b, j). This needs to be addressed.

Round 2

Reviewer 1 Report

The authors must continue to improve concepts such as the introduction, methodology, statistics, conclusions and updated references.

Author Response

We have updated the introduction by including updated references and edited the statistics methodology to include ANOVA analysis.